# Mask-Refined R-CNN: A Network for Refining Object Details in Instance Segmentation

**DOI:** 10.3390/s20041010

**Published:** 2020-02-13

**Authors:** Yiqing Zhang, Jun Chu, Lu Leng, Jun Miao

**Affiliations:** 1Department of Key Laboratory of Jiangxi Province for Image Processing and Pattern Recognition, Nanchang Hangkong University, Nanchang 330063, China; 1716085212012@stu.nchu.edu.cn (Y.Z.); miaojun@nchu.edu.cn (J.M.); 2School of software, Nanchang Hangkong University, Nanchang 330063, China; 3School of Electrical and Electronic Engineering, College of Engineering, Yonsei University, Seoul 120749, Korea; 4School of Aeronautical Manufacturing Engineering, Nanchang Hangkong University, Nanchang 330063, China

**Keywords:** instance segmentation, multi-scale feature fusion, Mask-Refined R-CNN, ROIAlign adjustment

## Abstract

With the rapid development of flexible vision sensors and visual sensor networks, computer vision tasks, such as object detection and tracking, are entering a new phase. Accordingly, the more challenging comprehensive task, including instance segmentation, can develop rapidly. Most state-of-the-art network frameworks, for instance, segmentation, are based on Mask R-CNN (mask region-convolutional neural network). However, the experimental results confirm that Mask R-CNN does not always successfully predict instance details. The scale-invariant fully convolutional network structure of Mask R-CNN ignores the difference in spatial information between receptive fields of different sizes. A large-scale receptive field focuses more on detailed information, whereas a small-scale receptive field focuses more on semantic information. So the network cannot consider the relationship between the pixels at the object edge, and these pixels will be misclassified. To overcome this problem, Mask-Refined R-CNN (MR R-CNN) is proposed, in which the stride of ROIAlign (region of interest align) is adjusted. In addition, the original fully convolutional layer is replaced with a new semantic segmentation layer that realizes feature fusion by constructing a feature pyramid network and summing the forward and backward transmissions of feature maps of the same resolution. The segmentation accuracy is substantially improved by combining the feature layers that focus on the global and detailed information. The experimental results on the COCO (Common Objects in Context) and Cityscapes datasets demonstrate that the segmentation accuracy of MR R-CNN is about 2% higher than that of Mask R-CNN using the same backbone. The average precision of large instances reaches 56.6%, which is higher than those of all state-of-the-art methods. In addition, the proposed method requires low time cost and is easily implemented. The experiments on the Cityscapes dataset also prove that the proposed method has great generalization ability.

## 1. Introduction

Thanks to the rapid development of flexible vision sensors and visual sensor networks, computer vision has entered a new phase. The improvements in computer vision-derived applications (such as vehicle tracking [1], facial interactions [2] and age assessment [3]) and various coding standards [4] also feed back the development of vision sensors. Vision sensors need the support of underlying algorithms, such as instance segmentation [5,6], image classification [7,8,9], object localization [10,11,12,13], and semantic segmentation [14,15,16,17]. For sensors that use instance segmentation as the underlying algorithm, such as sensors for autonomous driving and 3D vision, refined segmentation helps improve the safety of driving, the effect of 3D reconstruction, etc. Therefore, it is very important to refine object details in instance segmentation.

Instance segmentation (also called object segmentation) involves object detection and semantic segmentation [18]. Object detection [19] is commonly the first step in instance segmentation. The current instance segmentation model with satisfactory results can be regarded as an extension of two-stage detection, in which Faster R-CNN (faster region-convolutional neural network) [11] and FPN (feature pyramid network) [20] are the basis. Faster R-CNN provides a network foundation with high detection accuracy, and FPN is a common method combining layer features to improve the network performance. Mask R-CNN (mask region-convolutional neural network) [21] is based on Faster R-CNN and FPN. Mask R-CNN selects ROIAlign (region of interest align) layer instead of ROIPooling (region of interest pooling) layer, and adds a mask head onto the Faster R-CNN. To improve the efficiency, Mask R-CNN uses simple network structures in all parts of the network; a highly simple fully convolutional network is selected on the mask head. The objective is to reduce the computational cost while maintaining the segmentation accuracy. However, the framework only uses convolutional layers, batch normalization and deconvolutional layers, while the sizes of the feature maps do not change, in other words, the network neglects the details. Hence the masks predicted by Mask R-CNN are typically too smooth (as shown in Figure 1). To overcome this problem, a parallel fully connected layer path was inserted on the original fully convolutional mask head in the PAN (path aggregation network) [22], and the outputs of the two paths were merged. MS R-CNN (mask scoring region-convolutional neural network) [23] added a scoring path to the mask head to increase the supervision of the mask. The two network structures substantially improved the performance of Mask R-CNN; however, their segmentation accuracies were not good for large objects.

Mask-Refined R-CNN (MR R-CNN) is proposed in this paper for improving the segmentation accuracy, especially for large-scale objects. MR R-CNN is trained on the COCO (Common Objects in Context) dataset [24]. In contrast to the methods based on Mask R-CNN, the stride of ROIAlign is adjusted, and max pooling layers are added to the mask head to establish an FPN structure. This is the third time that the FPN structure has been selected, following the backbone and RPN (region proposal network) sections. In contrast to the first two FPN structures for object detection, the added FPN is used for semantic segmentation. This approach allows the network to consider the characteristics of various scales simultaneously and to improve the sensitivity of the network to the detailed information. Little computational cost is added by the modified mask head, and the overall network computing power is barely affected.

ROIAlign is the key operation of Mask R-CNN for Faster R-CNN, and the stride of ROIAlign is modified in our experiments. The quantization loss is reduced by ROIAlign, and the precise spatial position is retained faithfully. This algorithm has a remarkable impact: it can increase the mask accuracy by 10% to 50%. Let strd denote the stride; then strd = 32 denotes that the height and width of the feature map are both reduced to 1/32 of the original length. Experimental results on Mask R-CNN have demonstrated that the accuracy of strd = 32 is approximately 0.6% higher than that of strd = 16. However, we risk the effect reduction when using strd = 16 ROIAlign as the input of the mask branch in order to construct a feature pyramid framework based on feature maps of various scales to merge the features of multiple layers. According to [20], shallower networks focus more on detailed information, whereas higher-level networks focus more on semantic information. The integration of detailed information and semantic information enables the network to focus on more information. Fortunately, the accuracy improvement is more remarkable than the loss. Hence, a lateral connection is added to the refinement structure to suppress the amount of information of the feature maps, which prevents the forward-propagated signal from flooding the back-propagated signal.

As a summary, we propose MR R-CNN. Firstly, we build a feature pyramid for segmentation by establishing a refinement framework on a mask head and adjusting the stride of ROIAlign accordingly. In order to obtain higher segmentation accuracy, we also add lateral connection to the refinement framework to balance the amount of information. Secondly, we determine the optimal design scheme by adjusting the size of the input image, the number of feature fusions operations, and the means of feature fusion. In addition, we discuss the impact of using specific datasets on training models.

This paper is organized as follows: related works are reviewed in Section 2. Section 3 introduces our method, including the details of the pipeline and the entire algorithm. Section 4 introduces the experiments and analyzes the results. We make a concluding remark in Section 5.

## 2. Related Works

In this section, a brief overview of recent works on instance segmentation and multi-feature fusion are presented.

### 2.1. Instance Segmentation

The current methods for instance segmentation can be categorized briefly in two classes: the first class of instance segmentation methods is mainly based on image segmentation. Some works study specially designed transformations [25,26,27] or instance boundaries [28]. The instance mask is decoded from the predicted encodes. The DIN (dynamically instantiated network) [29] combines the prediction results of object detection semantic segmentation. [30] and [31] use RNN (recurrent neural network) to present an instance in each time step. However, these methods do not yield satisfactory results in complex environments.

The other class is based on object detection [32,33]. The object proposals are extracted through a detection framework. such as Faster R-CNN and R-FCN (region-based fully convolutional network) [34], and then semantic segmentation is conducted on the proposals. These methods depend heavily on the detection performance, especially the accuracy of localization. The instances cannot be correctly segmented if they are not properly positioned. Table 1 is a detailed introduction on the state-of-the-art methods.

### 2.2. Multi-Feature Fusion

Fusion has complementary advantages [37,38,39]. For semantic segmentation, FAIR’s SharpMask [40] combines the features of different layers to refine the details. A similar approach is LRR (Laplacian pyramid reconstruction and refinement) [41]. The FCN (fully convolutional network) for semantic segmentation [15] and U-Net [42] use backpropagation and skip connections to refine the segmentation. The feature-fusion methods based on U-Net differ substantially from the feature integration methods based on FCN. U-Net splices the features together in the channel dimension to form “thicker” features, whereas FCN fuses the corresponding points through the add operation and does not form “thicker” features.

For object detection, the SSD (single shot multibox detector) [13] and the DSSD (deconvolutional single shot detector) [43] assign the proposal to the appropriate feature level for processing. Similarly, FPN uses a bottom-up path with a lateral connection for object detection.

## 3. Proposed Method

### 3.1. Motivation

Mask R-CNN is currently the most influential instance segmentation framework. For efficiency, a fully convolutional structure of 4 convolutional layers and 2 deconvolutional layers are used in the mask head in Mask R-CNN, without any pooling layers. Various scales of semantic information of feature maps are ignored by the fully convolution structure. Therefore, the prediction of the image details is not satisfactory. According to [44], higher-level neurons are critical to global information, whereas lower level neurons are more susceptible to the activation of local textures and patterns. Therefore, the FPN structure is added to the mask head; hence, a top-down path is constructed and the higher-level features are propagated down to enhance the segmentation performance.

Our framework further enhances the segmentation performance of the entire feature hierarchy by fusing lower-level details and higher-level semantic information. To this end, as illustrated in Figure 2, a propagation path from the bottom layer to the upper layer is established and, accordingly, a lateral connection performs add operations between the feature maps of the same resolution, thereby constructing the feature pyramid structure.

In Figure 2, Strd = 32 is the original unilateral zoom size. C refers to the number of object categories. The input map (resize to 1024 × 1024) is fed to the backbone, and ROIAlign generates several fixed-size ROIs via the RPN and ROI functions. The generated ROIs are input into the three head networks of the classification, box, and mask, respectively. The maps (with size of 28 × 28) when strd = 16 for ROIAlign are transferred to the mask head, passing through three forward convolutional layers, and each layer is followed by a max pooling and a batch norm layer. Then, the feature map is zoomed in via deconvolution and added to the feature maps of the same size that are generated via the convolution with rectified linear unit (ReLU) in the forward direction and, finally, restore to the original resolution (28 × 28).

### 3.2. Mask-Refined Region-Convolutional Neural Network (MR R-CNN)

Mask R-CNN requires the correctness of the lightweight mask head. However, the “weight gain” of the mask head via a suitable method still has a substantial impact. The structure of MR R-CNN is illustrated in Figure 2. The details of our framework are as follows.

Mask R-CNN is the basic network of this article, and its pipeline is shown in Figure 3. Here is a brief introduction to this pipeline.

**Backbone:** For each input image, Mask R-CNN uses Residual Network (ResNet) as the backbone network for feature extraction. FPN is added to the backbone, which includes three channels: bottom-up, top-down and lateral connection. The bottom-up channel uses ResNet, which is divided into 5 stages according to the size of the feature map; except that stage1’s conv1 is not used, and the outputs of the last layer of stage 2 to stage 5 are defined as $C_{2},C_{3},\;C_{4},C_{5}$, respectively. Their stride to the original image is {4, 8, 16, 32}. The top-down channel is up-sampling from the highest layer, and the lateral connection is used to fuse the up-sampling results of the feature map of the same size generated from the bottom-up channel. Each layer in $C_{2},C_{3},\;C_{4},C_{5}$ undergoes a conv 1 × 1 operation, and all the output channels are set to 256, and then concatenate is operated with the up-sampled feature map.

**RPN:** After backbone, each feature image is input to RPN. RPN is divided into two paths. The first path classifies anchors through softmax to obtain positive and negative classification. The second parallel path calculates the bounding box regression offset of each anchor to obtain an accurate proposal. The final proposal layer is responsible for synthesizing positive anchors and corresponding bounding box regression offsets to obtain proposals, while excluding too small and out-of-boundary proposals. It is worth noting that the RPN will select the most appropriate scale from the output results of the backbone network feature pyramid $P_{2},P_{3},\;P_{4},P_{5}$ (for $C_{2},C_{3},\;C_{4},C_{5}$) to extract the region of interest (ROI). A formula to decide which $P_{k}$ the ROI of width w and height h should be cut from is:(1)$${k=⎣k}_{0}{+\log}_{2}(\frac{\sqrt{\mathrm{wh}}}{224})⎦$$

Here, 224 represents the size of images in ImageNet for pre-training. ROI with an area of w×h=224×224 should be at $k_{0}$-th level. Here, $k_{0}$ is 4, the optimized value experimentally obtained in Mask R-CNN, which means that the ROI of w×h=224×224 should be extracted from $P_{4}$. Assuming the scale of the ROI is less than 224 (112×112 for instance), ${k=k}_{0}-1=4-1=3$, this means that ROI will be extracted from higher-resolution $P_{3}$. This approach is reasonable. Large-scale ROIs should be extracted from low-resolution feature maps, which are good at detecting large objects, and small-scale ROIs should be extracted from high-resolution feature maps, which are good at detecting small objects. After this series of operations, the object is finally positioned.

**ROIAlign:** Mask R-CNN proposes ROIAlign to replace the ROIPooling in Faster R-CNN. In ROIPooling, there are two rounding processes. One is that {x, y, w, h} of each region proposal is usually not an integer, but it will be integerized for convenience. The second is to divide the integerized boundary area into k×k cells on average, and integerize the boundary of each cell. After the two roundings mentioned above, each region proposal at this time has deviated from the original position. To solve this problem, ROIAlign cancels the rounding operation and retains the decimals. The bilinear interpolation method is used to obtain float coordinates.

**Classification and bounding box regression:** As with Faster R-CNN, Mask R-CNN uses softmax to classify each acquired ROI, and the other parallel path performs bounding box regression and non-maximum suppression is performed in it to remove the bounding boxes marked multiple times for the identical object.

**Mask:** A new head network. Feature maps are classified at pixel level through a simple fully convolutional network consisting of convolutional layers and deconvolutional layers. Since each proposal contains only one foreground object, semantic segmentation of each proposal is equivalent to instance segmentation of the original image.

The segmentation mask of Mask R-CNN is not accurate enough. For semantic segmentation, improving the quality of the mask is always a challenge. This is because the receptive fields corresponding to adjacent pixels in the image often have very similar image information. This “similarity” has both advantages and disadvantages. If adjacent pixels are located inside the foreground object or background, then this “similarity” is advantageous, so these internal pixels can commonly be predicted correctly. However, if the adjacent pixels are located at the object edge, then this “similarity” will have a negative impact. The network structure of the mask head lacks the consideration of the receptive field. This causes the network to consider the global context information incompletely, and the relationship between pixels is also ignored.

In order to solve this problem, a feature pyramid network is established in the mask head in this paper. By fusing the information of feature maps of different scales, the network simultaneously takes into account the feature information of receptive fields of different sizes, and the network has sufficient contextual information to assist classification during segmentation. In order to achieve the goal of feature fusion, we must study the structure of the mask head network and adjust the scale of the input image (transmitted from ROIAlign) accordingly.

The essence of ROIAlign is “resize”. Its role is to convert a large number of feature images of different scales to the same size, thereby facilitating subsequent head network operations. The size of the image after ROIAlign is worth studying. Because the number of feature maps is much larger than the original image, if the size of the feature maps is too large, the subsequent head network will be overwhelmed and time-consuming. Although a small feature map scale is helpful to improve the prediction speed of the network, it will also degrade the prediction accuracy because of too much image information loss.

In Mask R-CNN, the maps transferred to ROIAlign with strd = 32, which is the optimized stride according to experiments. For mask prediction, this improvement is substantial. The creation of feature pyramids requires scaling the map multiple times and fusing the feature maps of various scales [20]. To create a feature pyramid in the mask head, it is not sufficient to rely on the original input size, and the model can only perform the add operation once. It is feasible to preserve the stride of ROIAlign and to magnify the map via bilinear interpolation in the mask head; however, the interpolation algorithm will damage the map information. Therefore, we keep the other head input sizes constant, multiply the stride of ROIAlign with the mask head, and neutralize the effect of this operation via the enhancement of the feature pyramid. Strd = 16 for ROIAlign because it is found to yield high accuracy.

The refinement framework is the core of the proposed algorithm. To obtain the best experimental results, 3 factors must be considered simultaneously, including the size of the image passed into the mask head network, the number of feature fusions operations, and the means of feature fusion. The size of the input image greatly affects the computation time; the number of feature fusions operation affects the degree to which the network considers the relationship between pixels and contextual information; and the way the feature fusion affects both the calculation speed and the segmentation accuracy. The optimized refinement frameworks finally arrived at are described below.

As shown in Figure 2, the feature map that passes through the ROIAlign of strd = 16 is input into our mask head, and it is doubled down by the forward network convolution-pooling-batch normalization operations until the size of the resulting feature maps is 7 × 7. The feature maps can no longer continue to be zoomed out. Then, through the reverse network, the map is enlarged by using 2 × 2 deconvolutional layers with ReLU, and the add operation is performed on the forward-propagating feature maps of the same resolution via a 3 × 3 convolutional layer and a ReLU. Next, the maps pass through a 3 × 3 convolutional layer and a ReLU. If the map in the forward network is directly transmitted, the amount of information will be large, and the back-propagated signal is likely to be flooded. Therefore, we reduce the amount of information of the forward maps via convolution and ReLU while maintaining the map resolution. The computational cost of the network is also decreased. After two operations, the map size is restored to 28 × 28, namely the size in [21], and the map is output by a 1×1 deconvolution. We have also tried to replace the add operation with the concatenate operation; its effect is poor compared with the add operation.

In order to further improve network performance, we conduct further research on the way of feature fusion. The fusion of feature maps that are propagated through the forward network and back-propagated maps is critical [20]. The amount of information of the feature maps from the forward network is much larger than that from the back-propagation network. The direct addition of these maps raises two issues: one is that the amount of computation is too large, and the other is that the former’s signal will flood the latter’s signal, thereby degrading the impact of the latter [45,46]. Therefore, a 3 × 3 convolutional layer with a ReLU is used to act on the forward network. This step just reduces the amount of information in the feature map and does not change the size of the feature map. Then, the feature maps are subjected to an add operation with the backpropagated map and, finally, are propagated forward through a 3 × 3 convolutional layer with a ReLU to reduce the amount of information. This lateral connection is highly effective, and the deeper the FPN, the higher the impact of this step.

The head of the network is trained through RPN proposals. The training samples must have an intersection over union (IoU) that is larger than 0.5 between the proposal and the corresponding ground truth. This is consistent with Mask R-CNN. To generate a regression object for each training sample, the prediction mask of the object class is obtained, and the prediction mask is binarized using a threshold size of 0.5. In training, the pre-training parameters are used. First, the backbone and ROI parameters are held constant, and the mask head is trained separately. After the optimal head has been determined, fine-tune training is conducted on the complete network.

## 4. Experiments and Results

### 4.1. Dataset and Evaluation Indices

MR R-CNN is trained on the COCO 2017 dataset [24]. COCO stands for Microsoft COCO: Common Objects in Context. It is a large dataset released by Microsoft in 2014, which can be used for computer vision tasks such as object detection, semantic segmentation, keypoint detection, and instance segmentation. The dataset aims at image analysis and understanding, and the image collection is derived from complex daily scenes. COCO is by far the largest instance-segmentation dataset, providing 80 object categories and more than 330,000 images, including 200,000 labeled images, and the total number of instances in the dataset exceeds 1.5 million.

The experiments are conducted in accordance with the COCO 2017 standard, with 115,000, 5000, and 20,000 samples in the training, validation and test sets, respectively.

Average precision (AP) is the evaluation index. As a basic evaluation indicator, IoU [47] is:(2)$$\mathrm{IoU}=\frac{\mathrm{area}\;\mathrm{of}\;\mathrm{overlap}}{\mathrm{area}\;\mathrm{of}\;\mathrm{union}}$$

True positive, false positive, true negative, and false negative are abbreviated as TP, FP, TN, and FN, respectively, accuracy [47] is:(3)$$\mathrm{accuracy}=\frac{\mathrm{TP}+\mathrm{TN}}{\mathrm{TP}+\mathrm{TN}+\mathrm{FP}+\mathrm{FN}}$$

AP refers to the average accuracy at 10 different IoU levels and 80 categories. The AP is evaluated at an IoU threshold of 0.5 to 0.95 with the interval of 0.05. Then the average of the 10 measurements is taken as the final AP result. The prediction accuracy rates of AP_50_ and AP_75_ are at the IoU thresholds of 0.5 and 0.75, respectively. The prediction accuracy rates are also evaluated for small objects, AP*_S_* (area ≤ 32^2^); medium objects, AP*_M_* (32^2^ < area ≤ 96^2^); and large objects, AP*_L_* (area > 96^2^).

### 4.2. Implementation Details

MR R-CNN is established on Mask R-CNN under TensorFlow. The pre-training ResNet [9] models are publicly available. MR R-CNN is compared with several state-of-the-art methods using Faster R-CNN/FPN/DCN (deformable convolutional networks)-FPN [48] based on ResNet-50/101. The size of input images is limited to a minimum of 800 px along the short axis and 1024 px along the long axis for training and testing. The official pre-training parameters are used, and the learning rate is set to 0.001. The experiment revealed that small learning rate is conducive to rapid convergence, which is related to the characteristics of the TensorFlow optimizer. An SGD (stochastic gradient descent) with a momentum of 0.9 is used as the optimizer. The number of ROIs used to train the model extracted for each image is set to a maximum of 200, and the ratio of positive and negative sample ROIs during training is set to 1: 3. Soft-NMS (non-maximum suppression) is also employed, and the top 100 score detections for each image are retained for testing.

### 4.3. Quantitative Results

MR R-CNN and Mask R-CNN are compared in Figure 1. In Table 2, MR R-CNN is compared with the state-of-the-art instance segmentation models: Mask R-CNN, FCIS, PAN and MS R-CNN. MR R-CNN substantially outperforms FCIS, which was the winner of the COCO 2016 Example Split Challenge. It outperforms Mask R-CNN using the same backbone in almost all indicators. MR R-CNN based on ResNet-101 DCN-FPN outperforms PAN based on ResNet-50 FPN. Compared with PAN, our model is lightweight, and the prediction time is shorter, because the data exchange frequently inside the time-consuming PAN. Our model outperforms MS R-CNN in terms of both AP*_M_* and AP*_L_*. Our model is sensitive to the details of objects; hence, the larger the object, the more details, and the stronger the the model.

To further evaluate the segmentation accuracy and generalization capability of MR R-CNN, the following experiments have been conducted. Our approach is compared with state-of-the-art methods in Figure 4. MR R-CNN outperforms them in most large instances. These images were all collected from the Internet, and the objects in the images have larger scales. Our approach slightly outperforms PAN and MS R-CNN. The misclassified areas of PAN and MS R-CNN have been manually outlined in the figure.

We manually selected 300 images from the Cityscapes [49] dataset for testing, in which the object sizes are relatively large. According to Table 3, MR R-CNN outperforms the Mask R-CNN baseline. MR R-CNN slightly outperforms PAN and MS R-CNN. The visual results are shown in Figure 5. The difference in visualization is not readily observed because the objects in the Cityscapes dataset are small. In summary, MR R-CNN has excellent segmentation accuracy for large objects and satisfactory generalization capability.

Our method is compared with the Mask R-CNN baseline in Table 4. Although the mask head becomes heavier, the computational cost is not increased substantially. The testing results on an NVIDIA 1080 Ti demonstrate that the average prediction time for a single image is increased by approximately 45 ms.

### 4.4. Ablation Study

ResNet-101 FPN is used as a backbone for an ablation study to train our network on the COCO 2017 dataset.

1. Refinement Framework

Six different refinement designs are shown in Figure 6.

Design (a) yields the best results among the 6 refinement designs with an ROIAlign mask head input of strd = 16.

Design (b) does not use the lateral connection. It yields the second-best results. However, since the amount of information in the forward network floods the amount of information that is propagated backward, it performs slightly worse than (a).

In Design (c), the input of ROIAlign to the mask head is kept unchanged. The feature maps of the two sizes are added by only a max pooling and deconvolutional layer. Then, the maps are deconvolved and output. Although this reduces the initial loss, it does not perform well due to insufficient information for fusion. 

Design (d) uses ROIAlign with strd = 8. Then, the feature pyramid structure is constructed, and a higher resolution map is output. Design (d) violates the original intention of the simple design of Mask R-CNN. It is difficult to train due to the excessive amount of data; hence, it is not successfully trained under our configuration.

Design (e) replaces the add operation in Design (a) with the concatenate operation. The result is also improved, but the result is not as accurate as that of Design (a). The training time under this design is increased substantially. The add operation blends low-level features, whereas the concatenate operation does not.

In Design (f), the original-sized maps are transferred into the mask head. Then, the length and width of the maps are doubled by bilinear interpolation. However, bilinear interpolation can severely damage the map, and substantially reduce the final accuracy.

The results of the above designs are presented in Table 5. The experimental results demonstrate that the characterization segmentation accuracy of the model is effectively improved by constructing the feature pyramid by ROIAlign with strd = 16 and the lateral connection.

2. Training Object

The COCO dataset contains many complex maps that cover multiple scales and occlusions of objects, which are challenging for instance segmentation tasks. Special images, including those containing only a single instance or only large, medium, or small instances, are used to train our model, to investigate whether the network can be more “focused” and learn more detailed features.

The qualified images in the COCO 2017 dataset are automatically selected by our edited scripts and converted into new training datasets. In addition to MR R-CNN, 4 additional training objectives are employed.

The impacts of various training objects are evaluated on the dataset *val* in the COCO 2017 dataset. The experimental results are presented in Table 6. Unfortunately, the selection methods for the above training objectives do not improve the segmentation accuracy of the network: only the result of (e) is close to that of (a). The reason for (b)’s deterioration is that training only on images with a single object can cause the network to focus on the object while ignoring the semantic information, such as occlusion and scale changes between objects. This can lead to poor network prediction in high spatial-structure complexity. Furthermore, in this training method, the number of training samples is much smaller compared to the previous training data, which results in class imbalances. Among (c), (d) and (e), only the results of (e) are close to those of (a). In (e), the prediction of the instance details is improved, because the loss is slightly reduced, which is caused by the special dataset. Since the network learns more object characteristic information through large objects, the prediction results of (c), (d) and (e) improve step by step. In summary, transferring the original image to the network for fine-tuned training is a satisfactory solution.

## 5. Conclusions

In this paper, we propose MR R-CNN to learn the effect of semantic segmentation of high-level and low-level features on instance segmentation. By adjusting the stride of ROIAlign, the FPN structure is added into the mask head, and the appropriate lateral connection structure is established; hence, the mask prediction is more sensitive to the instance details. Our method is easy to implement and performs well, while the speed is only slightly degraded. The experimental results on the COCO 2017 dataset and the Cityscapes dataset demonstrate that MR R-CNN outperforms the state-of-the-art methods based on Mask R-CNN. Our model performs remarkably well in the prediction of large objects, especially for details. However, our model is still constrained by localization accuracy. We will continue to modify our network to improve the localization accuracy in future work. 

## Figures and Tables

**Figure 1 sensors-20-01010-f001:**
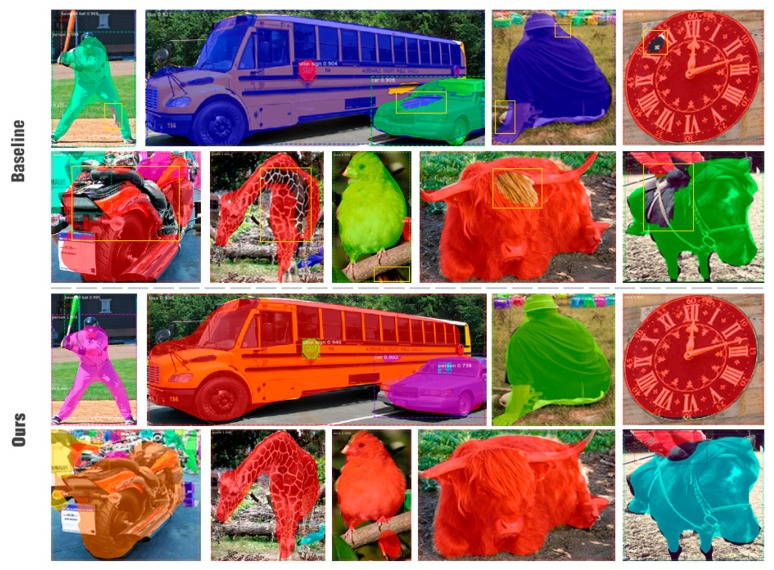
Mask R-CNN (mask region-convolutional neural network) baseline (top) vs. MR R-CNN (Mask-Refined R-CNN, bottom). The sample images are all screenshots of proposals. Our work can better segment large instances.

**Figure 2 sensors-20-01010-f002:**
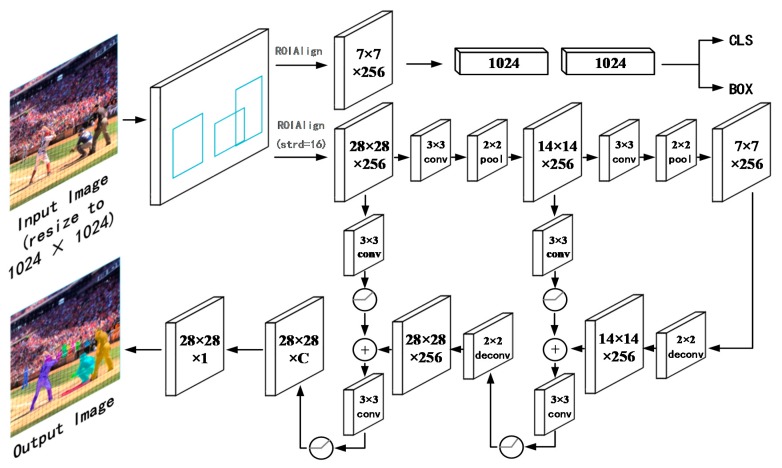
Network architecture of MR R-CNN.

**Figure 3 sensors-20-01010-f003:**
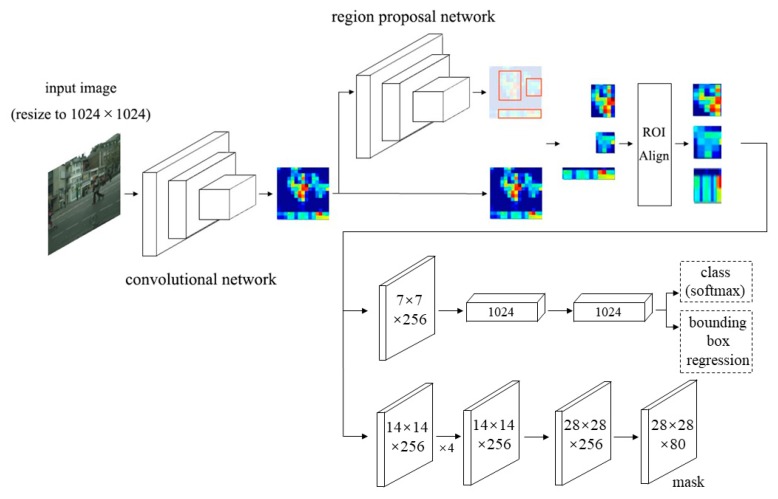
The pipeline of Mask R-CNN. ROIAlign and mask head are the key improvement points of Mask R-CNN over Faster R-CNN.

**Figure 4 sensors-20-01010-f004:**
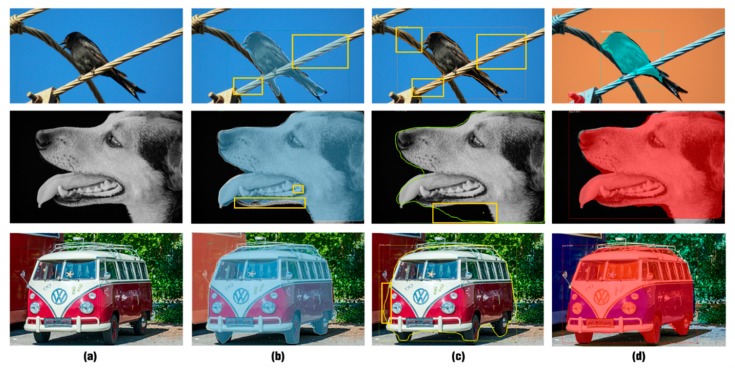
Comparison between MR R-CNN and state-of-the-art methods. (**a**) Original images, (**b**) PAN, (**c**) MS R-CNN and (**d**) MR R-CNN. Our approach performs better in most large instances.

**Figure 5 sensors-20-01010-f005:**
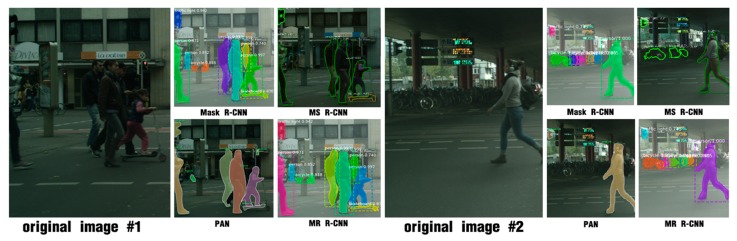
Comparison on the Cityscapes dataset between MR R-CNN and state-of-the-art methods.

**Figure 6 sensors-20-01010-f006:**
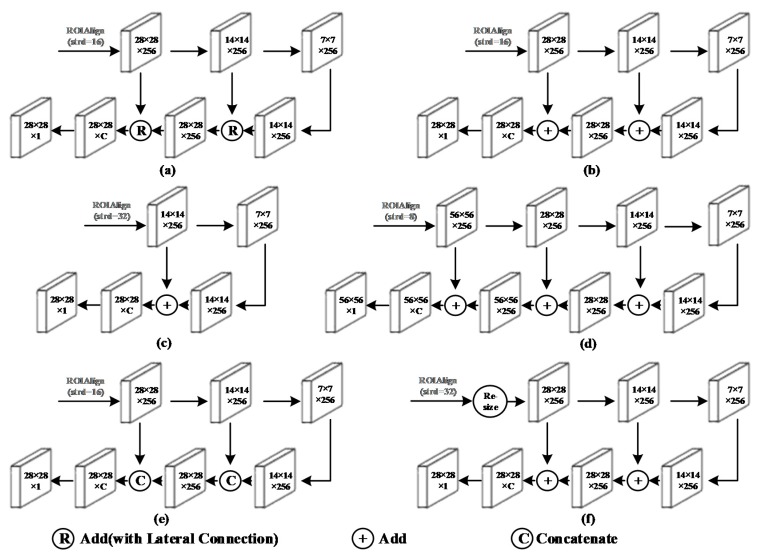
Designs for the input of the mask head and the refinement framework. Various network details are omitted for a more intuitive presentation.

**Table 1 sensors-20-01010-t001:** Detailed introduction on the state-of-the-art instance segmentation methods, including FCIS [35] (based on Instance-FCN [36]), Mask R-CNN, PAN and MS R-CNN.

Method	Year	Introduction	Shortcoming
FCIS	2017	The first end-to-end fully convolutional network instance segmentation framework. Added outside position-sensitive score maps (which implement classification and segmentation together) and RPN on Instance-FCN. Proposed instance-sensitive score maps to generate instance segmentation results.	Poor discrimination ability for overlapping objects.
Mask R-CNN	2017	The most influential method in instance segmentation. Added additional head for segmentation based on Faster R-CNN (extra segmentation head and original detection head do not share parameters). Changed the previous region of interest pooling (ROIPooling) to ROIAlign (region of interest align) using bilinear interpolation.	Weak ability to predict instance details.
PAN	2018	Proposed an additional feature pyramid network on Mask R-CNN. Improved the previous pooling strategy using adaptive feature pooling. Added a fully connected branch to the mask head, which greatly improves the prediction result.	High time cost.
MS R-CNN	2019	Added a scoring path prediction mask to the segmentation branch of Mask R-CNN. Added the gap between the prediction mask and the ground truth to the loss function, and obtained higher prediction accuracy.	Low accuracy for large instance.

**Table 2 sensors-20-01010-t002:** Instance segmentation mask on COCO (Common Objects in Context) test-dev. FCIS was the winner of COCO 2016. FCIS+++ uses more mature technology. Mask R-CNN’s network segmentation accuracy has been substantially improved, and it is the benchmark for the 2017 segmentation network model. PAN, proposed in 2018, yielded excellent results; however, the network is complex, the amount of data is large, and it requires more time. MS R-CNN in 2019 yielded the best results. MR R-CNN is suitable for predicting large and medium-sized objects, while other indicators are close to the best value.

Method	Backbone	AP	AP_50_	AP_75_	AP*_S_*	AP*_M_*	AP*_L_*
FCIS	ResNet-101	29.2	49.5	-	7.1	31.3	50.0
FCIS+++	ResNet-101	33.6	54.5	-	-	-	-
Mask R-CNN	ResNet-101-C4	33.1	54.9	34.8	12.1	35.6	51.1
Mask R-CNN	ResNet-101 FPN	35.7	58.0	37.8	15.5	38.1	52.4
Mask R-CNN	ResNeXt-101 FPN	37.1	60.0	39.4	16.9	39.9	53.5
PAN	ResNet-50 FPN	38.2	60.2	41.4	**19.1**	41.1	52.6
MS R-CNN	ResNet-101	35.4	54.9	38.1	13.7	37.6	53.3
MS R-CNN	ResNet-101 FPN	38.3	58.8	41.5	17.8	40.4	54.4
MS R-CNN	ResNet-101 DCN-FPN	**39.6**	**60.7**	**43.1**	18.8	41.5	56.2
MR R-CNN	ResNet-50 FPN	35.2	53.5	39.8	13.9	38.1	52.6
MR R-CNN	ResNet-101 FPN	37.6	56.1	41.1	16.4	40.6	54.7
MR R-CNN	ResNet-101 DCN-FPN	38.8	58.0	42.7	17.2	**41.8**	**56.6**

**Table 3 sensors-20-01010-t003:** Instance segmentation mask prediction results on 300 images from Cityscapes.

Method	AP	AP_50_	AP_75_
Mask R-CNN ResNet-101 FPN	34.2	53.6	36.6
PAN ResNet-50 FPN	37.5	56.1	40.6
MS R-CNN ResNet-101 FPN	37.8	56.5	41.0
MR R-CNN ResNet-101 FPN	**38.2**	**56.7**	**41.6**

**Table 4 sensors-20-01010-t004:** Average prediction time between the Mask R-CNN baseline and our network.

Method	Average Prediction Time
Mask R-CNN baseline	0.783s
MR R-CNN	0.828s

**Table 5 sensors-20-01010-t005:** Results of six different designs of the refinement framework.

Framework	AP	AP_50_	AP_75_
(a) MR R-CNN (with LC)	37.6	56.1	41.1
(b) MR R-CNN (without LC)	37.3	55.3	41.0
(c) strd=32 + Add	30.1	46.8	36.5
(d) strd=8 + Add	-	-	-
(e) strd=16 + Concatenate	33.4	53.3	37.2
(f) strd=32 + resize + Add	15.5	20.8	22.0

**Table 6 sensors-20-01010-t006:** Results of five choices of the training object.

Training Object	AP	AP_50_	AP_75_
(a) Original image	37.6	56.1	41.1
(b) Only one object	37.0	53.6	42.4
(c) Only small object (area ≤32^2^)	20.5	34.8	25.3
(d) Only medium object (32^2^ < area ≤ 96^2^)	32.2	51.9	36.1
(e) Only large object (area > 96^2^)	37.3	56.3	41.0

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
