# Peer review of "Mask-Refined R-CNN: A Network for Refining Object Details in Instance Segmentation"

_sensors, 2020, doi:10.3390/s20041010_

Round 1

Reviewer 1 Report

To overcome the available Mask R-CNN which the global context information is not fully considered, Mask Refined R-CNN (mask refined region-convolutional neural network, MR R-CNN for short) is proposed, in which the stride of ROIAlign is adjusted. The paper is well-written and addressed an important problem in the segmentation field, however, some important information is missing and a revision is required. Please consider bellow comments to improve the quality of your manuscript. 

The research gap, which is the problem of available Mask R-CNN is not well defined in the abstract. you mentioned that "Mask R-CNN does not always successfully predict instance details" which is because "the design of the semantic segmentation head network of Mask R-CNN is flawed, so the global context information is not fully considered". The second statement is not clear, why the global information can affect the prediction of the instant details. Please briefly discuss how the segmentation accuracy is measured. Please use the complete term for "AP" in the abstract. In the introduction, please discuss the issues of methods describe, and the need for your method. Please include some recent deep network works such as this paper, "deep convolutional neural network designed for age assessment based on orthopantomography data". Please include the input size (first layer) in Figure 2. In Figure 3, please includes the network size (similar to Figure 2), moreover, the text is hardly visible. Please discuss why K0 is set to 4 (Equation 1) and what is the effect of different values on segmentation results. Include references for datasets (for example in early section 4). Discuss the dataset size, the number of images, size of images and etc. Provide a reference for Eq. 2 and 3. 

Reviewer 2 Report

1. In my opinion, the novelty of this paper is somewhat marginal, so the authors need to describe the main contributions of this paper clearly in the Introduction section. 2. It is not clear how to balance the positive/negative training sample ratio? 3. The survey of related work is insufficient. The following relevant works are missing: “Advanced deep-learning techniques for salient and category-specific object detection: a survey” and “Learning rotation-invariant and Fisher discriminative convolutional neural networks for object detection”. 4. Please further polish the English language for a better readability.

Reviewer 3 Report

The paper proposes a new method for instance segmentation when performing object detection on visual data, extending an existing method (Mask R-CNN).

Although the proposed method is promising, with results that outperform the existing methods, the paper still needs improvements before it can be considered for publications. 

I suggest the following major revisions:

1) The title of the article is too bad. It is hard to understand what is being proposed and what is the research area that is being covered. The title must tell the readers what is the focus of the paper.

2) In general, information of the research objetives and methods is not clear along the text. This problem is more critical in the abstract and introduction sections. Particularly, the Introduction is very confusing and authors should be more clear when defining their scopes.

3) Related works section is too short. It lacks better comparisons with other works and a more detailed presentation of the state-of-the-art. A table comparing works would bring quality to the paper and proposed method. 

Although the results are valuable, authors should make a more detailed qualitative analyses of the method and the applicability in real applications (for example, citing computer vision and visual sensing problems). This could improve the conclusion section, which is too simple and generic.

Round 2

Reviewer 1 Report

All the comments have been addressed. Please consider bellow minor changes to improve the quality of your manuscript.

In Figure 1, please provide a reference for the images, which proposal used to screenshot to capture the images. 

Expand Figure 2 caption to include more information about the figure, for example, what is CLS and BOX, activation function, and etc.

Provide references for each method in Tables such as  2, 5, and 6.

Do not include text as an image in Figures, use original text instead to avoid blurring the text.

Reviewer 2 Report

I have no further comments.

Reviewer 3 Report

The authors have addressed my comments, resulting in an improved version of the paper.

I believe it is now suitable for publication.